# Manifold curvature and Ehrenfest forces with a moving basis

Jessica F. K. Halliday[1] and Emilio Artacho[1,2,3]

[1]*Theory of Condensed Matter, Cavendish Laboratory, University of Cambridge,*
*J. J. Thomson Ave, Cambridge CB3 0HE, United Kingdom*
[2]*CIC Nanogune BRTA and DIPC, Tolosa Hiribidea 76, 20018 San Sebastian, Spain*
[3]*Ikerbasque, Basque Foundation for Science, 48011 Bilbao, Spain*
(Dated: November 16, 2021)

Known force terms arising in the Ehrenfest dynamics of quantum electrons and classical nuclei, due to a moving basis set for the former, can be understood in terms of the curvature of the manifold hosting the quantum states of the electronic subsystem. Namely, the velocity-dependent terms appearing in the Ehrenfest forces on the nuclei acquire a geometrical meaning in terms of the intrinsic curvature of the manifold, while Pulay terms relate to its extrinsic curvature.

## I. INTRODUCTION

In a recent paper [1] it was established how to re-formulate quantum-mechanical equations for situations in which the basis set and the spanned Hilbert space vary with external parameters such as nuclear positions. This is routinely encountered in electronic structure calculations using atom-centered basis functions, and where the nuclei move, that is, any first-principles calculation method in quantum chemistry or condensed matter and materials physics using atomic orbitals as basis sets. There are many such methods and software packages that are widely used in either or both communities (for a brief review and links to codes used in quantum chemistry see, e.g., Ref. [2]; for methods and programs using atomic orbitals in condensed matter see, e.g., Refs. [3–9]). We will restrict ourselves here to mean-field-like methods, such as Hartree-Fock or Kohn-Sham density-functional theory (DFT) [10], including their time-dependent versions [11], and we will therefore use single-particle language.

For adiabatic situations, it is an old and well-known problem, for which the relevant equations have long been established. The key consequence of the basis states moving with nuclei is the appearance of Pulay forces [12], which are extra terms in the adiabatic forces acting on the nuclei due to the moving basis. The generalisation to non-adiabatic problems was done two decades ago [13–15]. Differential geometry concepts were recently used to obtain a transparent formalism, which allowed better insights into the meaning of the extra terms appearing in the equations [1]. The use of the formalism was demonstrated in two examples, namely, the time-dependent Schrödinger equation and the adiabatic forces on nuclei.

In this paper we extend the formalism of Ref. [1] to the Ehrenfest forces in mixed, classical-nuclei / quantum-electrons dynamical calculations. We show that the extra force terms appearing beyond the Pulay forces consist of a term proportional to the Riemann curvature tensor of the fibre bundle and the nuclear velocities, and, therefore, explicitly non-adiabatic, plus a term depending on the connection, which is implicitly non-adiabatic. The Pulay forces themselves are shown to appear only for curved manifolds, including extrinsically curved.

## II. FORCES

### A. General formalism

For Ehrenfest dynamics, considering a system of quantum electrons and classical nuclei, and following Todorov [14], we start from the Lagrangian

$$L = \sum_n^{N_e} \langle \psi_n | i\hbar\, d_t - H | \psi_n \rangle + \sum_j^{3N_n} \frac{1}{2} M_j \dot{R}_j^2 - V_{\text{nuc}}(\{R_j\}) \quad (1)$$

defined for the $N_e$ wavefunctions $\psi_n$ of independent electrons (we will disregard spin hereafter), and for the $3N_n$ position components of $N_n$ nuclei in three dimensions, $R_j$, as dynamical variables. $M_j$ represent the nuclear masses; $j$ runs over all nuclear-position vector components, and, therefore, the mass associated to the three components of a given nucleus is the same. $V_{\text{nuc}}(\{R_j\})$ stands for the nucleus-nucleus repulsion. $d_t$ represents the time derivative, $\frac{d}{dt}$, indicating $|\dot{\psi}_n\rangle$ in the Lagrange sense as $d_t|\psi_n\rangle$ (to distinguish it from the Lagrange partial derivatives, $\partial_j = \frac{\partial}{\partial R_j}$ and $\frac{\partial}{\partial\langle\psi_n|}$), although $d_t$ is still the partial time derivative in the Schrödinger sense when referring to, for instance, $\psi_n(\mathbf{r}, t)$.

$H$ is the effective single-particle Hamiltonian for the electrons using a mean-field theory such as the Kohn-Sham version of time-dependent density-functional theory [11]. All the results of this work directly apply to that theory [14, 15].

The evolution of both $\psi_n$'s and $R_j$'s will be then defined by minimising the action $S = \int^t L dt'$. This evolution was shown to conserve total energy, total momentum and the orthonormality of the wavefunctions [14].

Defining the first term as

$$L_e = \sum_n^{N_e} \langle \psi_n | i\hbar\, d_t - H | \psi_n \rangle \,, \quad (2)$$

we express now the electronic wavefunctions in a finite, non-orthogonal, and evolving basis set, $\{|e_\mu, t\rangle, \mu = 1 \ldots \mathcal{N}\}$, in an evolving $\mathcal{N}$-dimensional Hilbert space $\Omega(t)$, always a subspace of the entire (ambient) Hilbert

space $\mathcal{H}$. $\Omega(t)$ at all times defines a $(\mathcal{N}+1)$-dimensional fibre bundle $\Xi_t$. In its natural representation [1], and summing over repeated indices, Eq. (2) becomes

$$L_e = i\hbar \, \psi_{n\mu} \, d_t \psi^\mu{}_n - \psi_{n\mu} H^\mu{}_\nu \psi^\nu{}_n \ ,$$

with

$$\psi^\mu{}_n = \langle e^\mu | \psi_n \rangle \ , \ \ \psi_{n\mu} = \langle \psi_n | e_\mu \rangle \ , \ \ H^\mu{}_\nu = \langle e^\mu | H | e_\nu \rangle \ .$$

The set $\{|e^\mu, t\rangle, \ \mu = 1 \ldots \mathcal{N}\}$ is the dual basis of $\{|e_\mu, t\rangle\}$, also a basis of $\Omega(t)$, satisfying $\langle e^\mu, t | e_\nu, t \rangle = \delta^\mu_\nu, \ \forall \mu, \nu$ at any time $t$. The symbol $d_t$ indicates the covariant time derivative in $\Xi_t$, defined as [1]

$$d_t \psi^\mu{}_n = d_t \psi^\mu{}_n + D^\mu{}_{\nu t} \psi^\nu{}_n \ , \tag{3}$$

where $D^\mu{}_{\nu t} = \langle e^\mu | d_t e_\nu \rangle$ gives the connection in the manifold (note the convention in the order of indices).

There is also the possibility of orthonormalising the basis set at each time by, for instance, a time-dependent Löwdin transformation from the original non-orthogonal basis. In that case, the formalism remains, but it would simplify with the vectors of the dual basis becoming identical with their direct-basis corresponding vectors, and the metric tensors becoming the identity matrix. The equations all stay as for the natural representation with no need of distinguishing upper/lower (contravariant/covariant) indices. Numerically, however, it would be less efficient, so we keep the general non-orthogonal formalism for generality.

## B. Derivation of the forces

For the electrons, the Euler-Lagrange equations for the wavefunctions give [14] the time-dependent Schrödinger equation in the natural representation [1],

$$H^\mu{}_\nu \psi^\nu{}_n = i\hbar \, d_t \psi^\mu{}_n \ . \tag{4}$$

For the evolution of the nuclear coordinates, the Euler-Lagrange equations on $R_j$ give

$$M_j \ddot{R}_j = -\partial_j V_{\text{nuc}}(\{R_l\}) + \partial_j L_e \ ,$$

the last term representing the Ehrenfest forces on the ions due to the electrons (we will not include the nucleus-nucleus repulsion into the Ehrenfest forces as defined here).

Let us assume henceforth that the time evolution of the basis is associated to the nuclear motion, such that $|e_\mu, t\rangle = |e_\mu, \{R_j(t)\}\rangle$. This assumption includes the most widely used moving bases, which are fixed-shape atomic orbitals $f(\mathbf{r})$ for one quantum particle in three-dimensional space ($\mathbf{r}$ is 3D position) moving with the nuclei as $f(\mathbf{r} - \mathbf{v}t)$, being $\mathbf{v}$ the instantaneous velocity of the nucleus a particular basis function moves with. It is not limited to that case, however: the shape can vary (and does not need to be atomic-like), as long as it depends on

atomic positions and not explicitly on time [16]. We then define the $(\mathcal{N}+3N_n)$-dimensional fibre bundle $\Xi_R$ defined by $\Omega(\{R_j\})$, as spanned by the basis $\{|e_\mu, \{R_j\}\rangle\}$. The covariant derivative in this $\Xi_R$ manifold is now

$$\eth_j \psi^\mu{}_n = \partial_j \psi^\mu{}_n + D^\mu{}_{\nu j} \psi^\nu{}_n \ , \tag{5}$$

with the corresponding connection, $D^\mu{}_{\nu j} = \langle e^\mu | \partial_j e_\nu \rangle$. Both manifolds are related by any trajectory given by $\{R_j(t)\}$, which implies $d_t = v_j \eth_j$, being $v_j$ the nuclear velocities [17].

$L_e$ being a scalar, $\partial_j L_e = \eth_j L_e$. We compute the Ehrenfest forces on the nuclei as

$$F_j = \eth_j L_e = \eth_j \left( i\hbar \, \psi_{n\mu} \, d_t \psi^\mu{}_n - \psi_{n\mu} H^\mu{}_\nu \psi^\nu{}_n \right) \ .$$

We just need two other key facts to proceed. Firstly, in the Euler-Lagrange equations, the $\psi$'s and the $R_j$'s are treated as independent variables, which in the present formalism translates into $\eth_j \psi_{n\mu} = \eth_j \psi^\nu{}_n = 0$.

Secondly, using the Riemann curvature tensor of the bundle,

$$\Theta^\mu{}_{j\nu k} \psi^\nu{}_n \equiv \eth_j \eth_k \psi^\mu{}_n - \eth_k \eth_j \psi^\mu{}_n \ , \tag{6}$$

and the fact that $d_t = v_k \eth_k$, the double derivative in the first term of $F_j$ becomes

$$\eth_j d_t \psi^\mu{}_n = v_k \left( \eth_k \eth_j \psi^\mu{}_n + \Theta^\mu{}_{j\nu k} \psi^\nu{}_n \right) = v_k \, \Theta^\mu{}_{j\nu k} \psi^\nu{}_n \ ,$$

where we have used the fact that $\eth_j \psi^\mu{}_n = 0$.

These velocity-dependent terms in the forces were amply discussed by Todorov [14]. To our knowledge, the fact that they are simply velocity times curvature was not known. The Ehrenfest forces can then be concisely written as

$$\boxed{F_j = - \psi_{n\mu} \left( \eth_j H^\mu{}_\nu - i\hbar \, v_k \Theta^\mu{}_{j\nu k} \right) \psi^\nu{}_n.} \tag{7}$$

By introducing the explicit expressions for the curvature $\Theta^\mu{}_{j\nu k}$ and for the covariant derivative of the Hamiltonian $\eth_j H^\mu{}_\nu$ (both in Ref. [1]), the resulting expression coincides with what obtained in previous works by means of differential calculus [14, 15]. The meaning is now, however, much clearer, the expression more transparent, and the derivation much more direct.

## C. Non-adiabatic terms

In adiabatic (Born-Oppenheimer) evolution, the atomic forces can be expressed as [1]

$$F_j^{BO} = - \psi_{n\mu} \, \partial_j H^\mu{}_\nu \, \psi^\nu{}_n \ , \tag{8}$$

which is a direct result of the Hellmann-Feynman theorem. The difference between Eqs. (7) and (8) gives the two non-adiabatic basis-related terms.

### 1. Explicitly non-adiabatic

Firstly, the curvature term, $\psi_{n\mu} i\hbar \, v_k \Theta^\mu_{\ j\nu k} \psi^\nu_{\ n}$, is explicitly non-adiabatic, scaling linearly with the velocity of the displacing basis functions, with the moving nuclei. In the adiabatic limit of atoms moving infinitely slowly, that force term vanishes with $v_j \to 0$. The curvature itself is still there in the adiabatic limit, and it will give rise to effects analogous to the geometric phases found in similar contexts [18, 19], but the force itself is strictly non-adiabatic.

A possible visualisation of this force would be that of a (generalised) centripetal force, which is suffered by the nuclei due to the electrons being forced to evolve within the curved manifold. A more canonical interpretation can be obtained from the close analogy of what is presented in this work and the theoretical framework of geometric phases in molecular and condensed matter physics (see [18, 19] and references therein), as was already noted in Ref. [1]. In particular, the connection $D^\mu_{\ \nu j}$ appearing in the covariant derivative of Eq. (5) very closely relates to the Berry (or Mead-Berry) connection, or gauge potential [18, 19], while the curvature of Eq. (6) relates to the corresponding gauge field (gauge-covariant field strength). In this last sense, the velocity-dependent term in the non-adiabatic force can be seen as a (generalised) Lorentz force for a charged particle in a magnetic field.

The fundamental difference should be kept in mind, however, between this paper's theory and that relating to the mentioned geometric phases: the latter refers to the evolution of the problem's solutions, whereas the former relates to the evolution of the basis set. As hinted in Ref. [1], the relation between possible non-trivial behaviours in both manifolds could be interesting. It is clear that a topologically non-trivial solution manifold can exist in a trivial basis manifold (as in the limit of the latter tending to $\mathcal{H}$). The question is how non-trivial basis manifolds affect the topology of the solutions manifold. To our knowledge it is still to be explored.

### 2. Implicitly non-adiabatic

The second term is the difference between the $\psi_{n\mu} \eth_j H^\mu_{\ \nu} \psi^\nu_{\ n}$ term of Eq. (7) and the $\psi_{n\mu} \partial_j H^\mu_{\ \nu} \psi^\nu_{\ n}$ term of Eq. (8). Remembering the expression for the covariant derivative of the tensor associated to an operator, $\eth_j H^\mu_{\ \nu} = \partial_j H^\mu_{\ \nu} + D^\mu_{\ \sigma j} H^\sigma_{\ \nu} - H^\mu_{\ \lambda} D^\lambda_{\ \nu j}$ [1], that second non-adiabatic term can be expressed as

$$\psi_{n\mu} \left( D^\mu_{\ \sigma j} H^\sigma_{\ \nu} - H^\mu_{\ \lambda} D^\lambda_{\ \nu j} \right) \psi^\nu_{\ n} \, . \tag{9}$$

This term is not explicitly vanishing with velocity, it is rather an implicit non-adiabatic term. This is seen from the Hellmann-Feynman theorem, whereby, in the adiabatic limit, $H^\mu_{\ \nu} \psi^\nu_{\ n} = E_n \psi^\mu_{\ n}$ and $\psi_{n\mu} H^\mu_{\ \nu} = E_n \psi_{n\mu}$.

Therefore

$$\psi_{n\mu} \left( D^\mu_{\ \sigma j} H^\sigma_{\ \nu} - H^\mu_{\ \lambda} D^\lambda_{\ \nu j} \right) \psi^\nu_{\ n} =$$
$$= \psi_{n\mu} D^\mu_{\ \sigma j} E_n \delta^\sigma_{\ \nu} \psi^\nu_{\ n} - E_n \, \psi_{n\mu} \delta^\mu_{\ \lambda} D^\lambda_{\ \nu j} \psi^\nu_{\ n}$$
$$= E_n \, \psi_{n\mu} \left( D^\mu_{\ \nu j} - D^\mu_{\ \nu j} \right) \psi^\nu_{\ n} = 0 \, .$$

However, if $\psi^\mu_{\ n}$ is not an eigenstate of $H^\mu_{\ \nu}$, i.e., it is evolving non-adiabatically, then the term in Eq. (9) is not zero. Explicitly, given the basis of Hamiltonian eigenstates at any time $\xi^\mu_a$ (that is, $H^\mu_{\ \nu} \xi^\nu_a = \varepsilon_a \xi^\mu_a$), if expanding the evolving $n$-th state as $\psi^\mu_{\ n} = \sum_a C_{an} \xi^\mu_a$, Eq. (9) becomes

$$\psi_{n\mu} \left( D^\mu_{\ \sigma j} H^\sigma_{\ \nu} - H^\mu_{\ \lambda} D^\lambda_{\ \nu j} \right) \psi^\nu_{\ n} =$$
$$= \sum_{a,b} C^*_{an} C_{bn} (E_b - E_a) \xi_{a\mu} D^\mu_{\ \nu j} \xi^\mu_b \, ,$$

This expression becomes zero when at most only one of the $C_{an}$ coefficients is non-zero, which is precisely the adiabatic case.

### D. Pulay forces

There is one last point to make in the relation between forces and the curvature of the manifold, which was already implicit in Ref. [1]. Pulay forces appear in the calculation of the matrix elements $\partial_i H^\mu_{\ \nu} = \partial_i \langle \psi^\mu | H | \psi_\nu \rangle$, as

$$\partial_i H^\mu_{\ \nu} = \langle e^\mu | \partial_i H | e_\nu \rangle + \langle \partial_i e^\mu | H | e_\nu \rangle + \langle e^\mu | H | \partial_i e_\nu \rangle \, , \tag{10}$$

the Pulay terms being the last two. From the expression for the covariant derivative of the Hamiltonian tensor, $\eth_i H^\mu_{\ \nu}$, Ref. [1] recast Eq. 10 in the more revealing form

$$\eth_i H^\mu_{\ \nu} = \langle e^\mu | \partial_i H | e_\nu \rangle + \langle \partial_i e^\mu | Q_\Omega H | e_\nu \rangle + \langle e^\mu | H Q_\Omega | \partial_i e_\nu \rangle,$$

where $Q_\Omega$ is the complement of the projector onto $\Omega \in \mathcal{H}$, $P_\Omega$, i.e., $P_\Omega + Q_\Omega = \mathbb{1}$, the identity operator in the infinite-dimensional ambient Hilbert space $\mathcal{H}$. The last two terms of the last expression make explicit the direct relation between the Pulay correction and the curvature of the manifold: if the extrinsically defined basis vectors stay within $\Omega$ when displacing coordinate $i$, as would happen in the absence of curvature, $Q_\Omega | \partial_i e_\nu \rangle = |0\rangle$ and $\langle \partial_i e^\mu | Q_\Omega = \langle 0 |$, and, therefore, the last two terms would be zero, giving $\eth_i H^\mu_{\ \nu} = \langle e^\mu | \partial_i H | e_\nu \rangle$. The effect of basis change within $\Omega$ is taken care of by the connection in the covariant derivative in $\eth_i H^\mu_{\ \nu}$ [inside parenthesis in Eq. (9)], which, as shown in the previous subsection, give a zero contribution to the forces in the adiabatic case. In other words, when slightly displacing in nuclear configuration space, it is not the change in basis, but the turning of $\Omega$, what matters for the (adiabatic) Pulay corrections to the forces, which happens when $\Xi_R$ is curved.

Unlike the velocity-dependent terms discussed above, which depend on the intrinsic (Riemann) curvature, the

Pulay corrections appear for any curvature, including extrinsic curvature (in the sense of a cylinder having non-zero extrinsic curvature but zero intrinsic one), since the Pulay corrections stem from calculations in the ambient space $\mathcal{H}$, including outside $\Omega$. The corrections will be there as long as $Q_\Omega|\partial_i e_\nu\rangle \neq |0\rangle$.

## III. CONCLUSION

For Ehrenfest dynamics of quantum electrons and classical nuclei, and for basis functions for the former that move with the latter, it has been shown how the extra terms appearing in the Ehrenfest forces acquire a natural geometric interpretation in the curved manifold given by the set of electronic (tangent) Hilbert spaces defined at each set of nuclear positions. The velocity-dependent term, explicitly non-adiabatic, depends on the intrinsic curvature of the manifold (it could be considered to be a centripetal force arising when constraining motion to the curved manifold, or the force arising due to the effective gauge field represented by that curvature). It has the simple form of velocity times curvature.

The two additional terms are implicitly non-adiabatic, disappearing in the adiabatic limit, when following the Born-Oppenheimer surface. The well-known Pulay forces are also shown to be a consequence of the manifold curvature, although in this case, an extrinsically curved manifold is enough for these terms to appear.

The paper allows a deeper understanding of the extra terms appearing in the Ehrenfest forces for moving basis sets, in addition to connecting them to other contexts, albeit the forces themselves are unchanged. This is unlike what happens with the better understanding of the electronic evolution equation, Eq. (4), which enables the design of better numerical integrators [20]. The curvature itself, however, Eq. (6), can also be exploited as a measure of basis incompleteness along a nuclear trajectory.

## ACKNOWLEDGMENTS

J. Halliday would like to acknowledge the EPSRC Centre for Doctoral Training in Computational Methods for Materials Science for funding under grant number EP/L015552/1. E. Artacho is grateful for discussions with Prof. Christos Tsagas, and acknowledges funding from the Leverhulme Trust, under Research Project Grant No. RPG-2018-254, from the EU through the ElectronStopping Grant Number 333813, within the Marie-Curie CIG program, and by the Research Executive Agency under the European Union's Horizon 2020 Research and Innovation programme (project ESC2RAD, grant agreement no. 776410). Funding from Spanish MINECO is also acknowledged, through grant FIS2015-64886-C5-1-P, and from Spanish MICINN through grant PID2019-107338RB-C61/ AEI /10.13039 / 501100011033.

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
