# Peer review of "Manifold curvature and Ehrenfest forces with a moving basis"

_SciPost Physics_

## Round 1 · Referee Report · Anonymous (Referee 1) · 2021-9-23

Report

The Authors of the Manuscript "Manifold curvature and Ehrenfest forces with a moving basis" present an enlightening geometric interpretation of the forces appearing in the Ehrenfest dynamics (ED).

I essentially agree with the Authors when they state that: "To our knowledge, the fact that they [the velocity-dependent terms in ED] are simply velocity times curvature was not known". On the other hand, there is some remarkable connection between their presentation and the "gauge theory of molecular physics" as presented, e.g., in Chapter 8 of A. Bohm et al. "The Geometric Phase in Quantum Systems", Springer Berlin Heidelberg, 2013. The connection $D^{\mu}_{\nu j}$ in the Manuscript seems related to what A. Bohm et al. call "Mead-Berry connection". This is a gauge field which appears as a (matrix) analogue of the vector potential after that the electronic degrees of freedom have been traced out. The second-last equation of the first column on page 2 of the Manuscript seems related to the kinematic momentum operator of the gauge theory of molecular physics. The Riemann curvature tensor defined in the first equation of the second column on page 2 is referred to as gauge-covariant field strength by A. Bohm et al. and it is the (matrix) analogue of a magnetic field. The velocity-dependent term of the forces from a moving basis seems then the analogue of a Lorentz force. Interestingly, the field strength of A. Bohm et al. is identically equal to zero if the trace over the electronic states extends over the entire Hilbert space. This is not necessarily true if a partial trace is taken instead, which A. Bohm et al. call the "Born-Huang" approximation. This approximation bears similarity to the analysis of the Pulay forces considered by the Authors at the end of the Manuscript.

The Authors may want to add a discussion of the similarities and differences between their formalism and the "gauge theory of molecular physics" of A. Bohm et al.

The following points only require some minor clarification:

  • In the Eq. (1), the extreme of the second summation may be $N_n$.

  • At the end of the second column on page 1, the Authors state that the basis set is made of non-orthogonal electronic states. Why is it not possible to orthogonalise the basis set at a time "t", given that the Hilbert space is finite-dimensional? Incidentally, it is not stated that the Hilbert space $\Omega(t)$ is a subspace of the entire Hilbert space. Is this the case?

  • In the first column of page 2, the Authors state that a fixed-shape moving orbital can be written as f(r-vt). I am not sure whether the symbol "r" has been previously introduced. Additionally, it is not clear whether "v" is a time-dependent (instantaneous) nuclear velocity or a constant velocity.

  • On the second column of page 2, the statement "In the adiabatic limit of atoms moving infinitely slowly..." seems to imply that there are no geometric effects in the adiabatic case. However, in the adiabatic case the curvature is responsible for the appearance of a geometric phase, e.g., in the case of a diatomic molecule (Phys. Rev. Lett., 56, 893, 1986) or in the so-called molecular Aharonov—Bohm effect (Chem. Phys. 49, 23, 1980).

  • On the second column of page 2, the sentence starting with "Remembering that ..." shows an identity which is not entirely trivial. A short explanation or a reference may help.

  • In the first paragraph of the first column of page 3, the basis $\xi^{\mu}_a$ should be defined in a clearer way.

  • In the same column, Section D, the definition of the projectors can be also clarified by an earlier statement that $\Omega$ is a subspace of the entire Hilbert space, if this is indeed the case.

  • I have found Section D quite terse. It is not immediately clear why the two equations shown there should be equivalent. It is also not immediately clear why the last two terms of the second equation are related to the Riemann curvature. The final comment "In other words, when slightly displacing in nuclear configuration space, it is not the change in basis..." does not clarify the previous discussion and should be extended. In particular, the difference between "intrinsic" and "extrinsic" effects is not entirely clear to me. I suppose "intrinsic" relates to geometric effects, while "extrinsic" to effects due to the "truncation" of the Hilbert space. On the other hand, the geometric effects are trivial --- in the sense that can be gauged away --- if the entire electronic Hilbert space is considered, unless the topology of the gauge field is not trivial, e.g., if there are conical intersections. This possibility is also briefly mentioned in Ref. [1] of the Manuscript.

  • validity: -
  • significance: -
  • originality: -
  • clarity: -
  • formatting: -
  • grammar: -

Author:  Emilio Artacho  on 2021-09-27  [id 1788]

(in reply to Report 1 on 2021-09-23)

We thank the person issuing the invited report for the careful reading of the paper and the insightful comments and suggestions. Yes, there is a clear connection to geometric phases, which was already briefly discussed in an earlier publication on the general topic (Reference 1 of the submitted manuscript), but, the referee is right, the connection was not made in this submission for the particular expressions in it, and it would be useful for the reader. There is a parallelism in formalism, although the geometric characterisation mentioned by the referee refers to the wavefunction(s) describing the state of the system, while the one in our work relates to the manifold given by the evolving basis set (as opposed to evolving eigenstates). We will add a brief discussion.

The further minor clarifications will also be added as requested, since they will indeed add clarity to the text. Thanks.

---

## Round 1 · Referee Report · Anonymous (Referee 2) · 2021-10-1

Report

Admittedly I'm not an expert in Riemann geometry. I just have two questions/remarks: - the velocities v_k are the time-derivative of the nuclei position R_k. Why not using a capital "V" then? - After Eq. (5), it's written that the term does not vanish with the zero velocity. I'm puzzled because I understand that the term D = <e^\mu | d_t e_\nu > should vanish if the basis function e_\nu does not move with the time. Could you please clarify?

  • validity: -
  • significance: -
  • originality: -
  • clarity: -
  • formatting: -
  • grammar: -

Author:  Emilio Artacho  on 2021-10-03  [id 1799]

(in reply to Report 2 on 2021-10-01)

We thank the second referee for the careful reading of the manuscript. The clarifications requested are helpful and will be useful for readers.

  • "the velocities v_k are the time-derivative of the nuclei position R_k. Why not using a capital "V" then?"

It is a nomenclature choice. Yes, it would seem more consistent and possibly transparent as the referee suggests. We just saw that lower case v is inambiguous (no clashes within the paper) while capital V was used for the potential in the Lagrangian.

  • "After Eq. (5), it's written that the term does not vanish with the zero velocity. I'm puzzled because I understand that the term D = <e^\mu | d_t e_\nu > should vanish if the basis function e_\nu does not move with the time. Could you please clarify?"

The D term discussed in Eq. (5) is <e^\mu | d_j e_\nu>, the derivative with respect to the position of nuclear coordinate j, there is no time derivative there. Actually, the D_t the referee mentions would become v_j D_j, and would indeed be velocity dependent. But the referee is probably referring to the expectation that, if the basis set does not move, one might not expect terms depending on such D_j derivatives, which happens to be right. There maybe more general ways to show that, but the one used after Eq. 5 is a simple one we could come up with.

Anonymous on 2021-10-04  [id 1801]

(in reply to Emilio Artacho on 2021-10-03 [id 1799])
Category:
answer to question

I apologize. This is my mistake.
There are 2 terms named "D", however with different indeces:
<e^\mu | d_t e_\nu> in Section II.A
<e^\mu | d_j e_\nu> in Section II.B
The paper is correct, even though the notations are a bit confusing here.

---

## Editorial Decision

resubmitted